# Effect of Capsaicin Addition on Antioxidant Capacity, Immune Performance and Upper Respiratory Microbiota in Nursing Calves

**DOI:** 10.3390/microorganisms11081903

**Published:** 2023-07-27

**Authors:** Minqiang Su, Yuanhang She, Ming Deng, Yongqing Guo, Yaokun Li, Guangbin Liu, Baoli Sun, Dewu Liu

**Affiliations:** 1College of Animal Science, South China Agricultural University, Guangzhou 510642, China; smq2054113415@163.com (M.S.); syh15521170780@163.com (Y.S.); dengming@scau.edu.cn (M.D.); yongqing@scau.edu.cn (Y.G.); ykli@scau.edu.cn (Y.L.); gbiiu@scau.edu.cn (G.L.); 2National Engineering Research Center for Breeding Swine Industry, College of Animal Science, South China Agricultural University, Guangzhou 510642, China; 3National Local Joint Engineering Research Center of Livestock and Poultry, South China Agricultural University, Guangzhou 510642, China; 4Collaborative Innovation Center for Healthy Sheep Breeding and Zoonoses Prevention and Control, Shihezi University, Shihezi 832000, China

**Keywords:** capsaicin, calves, immunity, upper respiratory tract microbiome

## Abstract

Capsaicin (CAP) has various biological activities; it has antibacterial, anti-inflammatory and antioxidant properties, and stimulates intestinal development. The aim of this study was to investigate the effect of CAP on the health of nursing calves under group housing conditions. Twenty-four newborn Holstein calves were randomly assigned to three treatment groups of eight calves each. The milk replacer was supplemented with 0, 0.15 or 0.3 mL/d of CAP in each of the three treatment groups. Following a one-month clinical trial of individual-pen housing, an extended one-month trial of group housing was conducted. At the end of the trial, serum samples, rectal fecal samples and upper respiratory swab samples were collected to determine the effect of CAP addition on serum parameters, fecal fermentation parameters and upper respiratory microbiota of calves under group housing conditions. The results showed that the addition of high doses of CAP decreased calf respiratory scores (*p* < 0.05), increased serum glutathione peroxidase, superoxide dismutase, immunoglobulin A, immunoglobulin G, immunoglobulin M and interleukin-10 concentration (*p* < 0.05), and decreased malondialdehyde, amyloid A and haptoglobin concentration (*p* < 0.05). Moreover, high doses of CAP increased the rectal fecal concentration of total short-chain fatty acids, acetate and butyric acid (*p* < 0.05). In addition, CAP regulated the upper respiratory tract microbiota, with high doses of CAP reducing *Mycoplasma* abundance (*p* < 0.05), two doses of CAP reducing *Corynebacterium* abundance (*p* < 0.05) and a tendency to reduce *Staphylococcus* abundance (*p* = 0.06). Thus, CAP can improve calf antioxidant capacity, immune capacity and reduce inflammatory factors, stress proteins as well as improve gut fermentation and upper respiratory microbiota under group housing conditions, which is beneficial for healthy calf growth.

## 1. Introduction

Calf production is associated with various stressors and health challenges that can impact their growth and overall well-being. These stressors include weaning, transportation, group housing, as well as changes in dietary practices and rearing environment [1]. Bovine respiratory disease (BRD) is a common and severe illness in calf production, leading to high mortality rates and significant economic losses, affecting both the upper respiratory tract (URT) and lower respiratory tract of cattle [2]. Group housing conditions increase the transmission rate of pathogens and the incidence of BRD [3,4]. Antibiotics have been widely used as an effective treatment for BRD in dairy production. However, the extensive use of antibiotics has resulted in the emergence of drug-resistant pathogens, posing a serious threat to the health and safety of livestock and humans [5]. The use of plant extracts as alternatives to antibiotics has become a research focus in the current livestock industry.

The red pepper plant of the family Solanaceae is extracted and processed to obtain a series of capsaicinoids, the most abundant of which is capsaicin (CAP), which accounts for 71% of the total capsaicinoid content in red pepper [6]. Capsaicin is a pungent vanilloid alkaloid that exhibits various biological functions, such as analgesic, antimicrobial, anti-inflammatory, antioxidant, anticancer, digestive, and metabolic regulatory activities [7,8]. Currently, research on capsaicin has primarily focused on monogastric animals and adult dairy cows. In rats, CAP has been found to enhance antioxidant capacity and inhibit oxidative stress [9]. In weaned piglets, CAP improves average daily gain (ADG), antioxidant capacity, digestive enzyme activity, and intestinal fermentation, while suppressing the expression of inflammatory factors [10]. In nursing dairy cows, CAP reduces milk production without affecting feed efficiency [11]. In beef cattle, CAP does not affect ADG and feed intake but increases serum antibody concentrations of bovine respiratory syncytial virus [12]. Based on the available literature, we hypothesized that CAP may positively affect the immune response of nursing calves under group housing conditions. Therefore, in order to evaluate the effect of CAP on the immunocompetence and upper respiratory microbiota of calves in a group housing environment, the present study was continued with a 1-month group housing clinical trial after a one-month clinical trial of individual-pen housing.

## 2. Materials and Methods

This study was conducted from November 2022 to January 2023 in a commercial dairy farm in Qingyuan City, Guangdong Province, China. The average temperature was 17.1 °C (5.8–31.4 °C) and the average humidity was 64.2% (14.8–85.7%) during the time the experiment was conducted. All animals used in this study conformed to the guidelines for the care and use of laboratory animals established by the Ministry of Agriculture, China, and were approved by the Ethics Committee of South China Agricultural University (SCAU, Guangzhou, China; approval number: 2021g019).

### 2.1. Experimental Design and Treatments

In our study, we randomly selected 24 newborn Chinese Holstein female calves to participate in this clinical experiment (BW = 36.67 ± 3.44 kg). The experimental calves were separated from their dams immediately after birth, weighed, and fed colostrum equivalent to 10% of their body weight. The bovine colostrum was thawed in a colostrum pasteurizer machine prior to feeding (>22% Brix). Calves were divided into 8 groups of 3 calves each according to birth order and birth weight, and 1 calf was randomly selected from each group and assigned to 3 groups: group C (no CAP added, n = 8), group L (1% CAP 0.15 mL/day/calf, n = 8) and group H (1% CAP 0.3 mL/day/calf, n = 8). Jugular vein blood samples were collected from calves at 3 days of age, and total serum protein concentration was determined using a portable refractometer; passive immunization was judged successful if it exceeded 5.5 g/dL. All calves in this study exceeded the threshold serum total protein concentration, averaging 6.7 ± 0.7 g/dL. Calves were started in a 1-month individual-pen housing clinical experiment at 5 days of age, and in order to investigate the effect of capsaicin on the health of nursing calves after a change in the housing environment, this was followed by a 1-month group housing clinical experiment until the end of the experiment at 66 ± 2.04 days of age. The data in this article contain clinical trial data for the group housing phase only. The capsaicin used was a water-soluble capsaicin with a concentration of 1%, with emulsifiers and water as additional components, produced by Leader Biotechnology Company, Guangzhou, China.

### 2.2. Housing, Management and Dietary Treatments

All test calves started the formal experiment at 5 days of age and were kept in individual pens (1.6 m long, 0.9 m wide, and 1.5 m high) from 5 days of age to 35 ± 2.04 days of age. In front of each pen, there were water and feed buckets for calves to drink and feed freely, and new buckets were changed every morning. The bedding was changed every 2 days using straw as bedding. Milk replacer (MR, 202 g/kg crude protein, 162 g/kg ether extract, dry matter basic, 160 g powder as feed/L, Nutrifeed, Netherlands) was bottle-fed at 7:00 and 16:00 daily, 10 L/d (5 L/meal). The reconstituted MR was heated to 42 °C before feeding. Before each feeding, 0.15 mL/d (0.075 mL/meal) or 0.3 mL/d (0.15 mL/meal) of CAP was added to 1.5 L of MR in the corresponding treatment group and mixed well before feeding. We ensured that all MR containing capsaicin was consumed before feeding the remaining MR. At 36 ± 2.04 days of age, calves were transferred from individual pens to group pens (30 m^2^) and 8 calves of the same group were fed in the same pen until 66 ± 2.04 days of age. The group pens were equipped with water and feed troughs for calves to drink freely and to feed on a concentrate and roughage diet with a 1:6 mixture of oat hay and calf starter, and the water and feed troughs were cleaned daily. We used 5 cm-thick wood bran as bedding and changed it once a week. MR was fed daily at 07:00 and 16:00 using a feeder with 8 teats, and MR was fed into the feeder at each meal at the total MR feeding rate of 8 calves. The appropriate dose of CAP (0.075 mL/meal or 0.15 mL/meal) was added to 1.5 L of MR before each feeding and bottle fed to the respective treatment group of calves to ensure that all were consumed. A total of 12 L/d MR was fed at 36–45 ± 2.04 days of age, after which the MR supply was reduced by 1 L/d for 10 d until weaning at 66 ± 2.04 days of age.

### 2.3. Feed Analysis

Feed samples, including roughage, starter feed, leftovers and milk replacer, were collected weekly and pooled for each period. The feed samples were dried at 105 °C for 48 h to determine dry matter (DM) content. The dried samples were ground using a grinder and passed through a 40-mesh sieve. The feed samples were incinerated in a muffle furnace at 550 °C for 4 h to determine the crude ash content [13]. The samples that had been ground and passed through the sieve were subjected to the Kjeldahl method to determine crude protein (CP) content [14]; the Soxhlet extraction method was used to determine ether extract (EE) content [13]. Acid detergent fiber (ADF) and neutral detergent fiber (NDF) were determined using the method described by Van Soest et al. [15], using anhydrous sodium sulfate and heat-stable α-amylase, with the use of a fiber analyzer. Non-fiber carbohydrates (NFC) in the feed samples were estimated according to Mertens [16]: NFC (%) = 100% − (NDF% + CP% + EE% + Ash%). The nutrient composition of the feed is shown in Table 1.

### 2.4. Health Exams

All calves were scored daily for the respiratory systems by the same trained veterinarian. Respiratory disease was considered to be present when at least 2 clinical parameters were scored abnormally according to McGuirk and Peek [17], which included nose, eye, ear, cough and rectal temperature scores. Antimicrobial treatment was administered only if the calves were febrile or depressed and if they reduced or refused to drink milk, using a single dose of florfenicol plus flunixin meglumine in a weight-based formulation (0.05 mL/kg florfenicol, 0.02 mL/kg flunixin meglumine).

### 2.5. Determination of Serum Parameters

On the final day of the experiment, approximately 10 mL of blood samples were collected from each calf via jugular vein puncture using a separator gel procoagulation tube. The blood samples were centrifuged at 2500 g/min for 15 min at 4 °C to obtain serum, which was stored at −20 °C prior to analysis. Serum glucose (Glu), alanine aminotransferase (ALT), aspartate aminotransferase (AST), alkaline phosphatase (ALP), total protein (TP), albumin (ALB), blood urea nitrogen (BUN), and creatinine (CR) were measured using an automated biochemistry analyzer (CLS880, Weifang, China). Serum immunoglobulin A (IgA), immunoglobulin G (IgG), and immunoglobulin M (IgM) were determined using ELISA kits (Beijing Kyushu Taikang Biotechnology Co., Ltd., Beijing, China). Serum interleukin-1β (IL-1β), interleukin-6 (IL-6), interleukin-10 (IL-10), tumor necrosis factor-α (TNF-α), serum amyloid A (SAA) and haptoglobin (HP) were measured using ELISA kits (Shanghai Zhangshi Biotech Co., Ltd., Shanghai, China). Total antioxidant capacity (T-AOC), malondialdehyde (MDA), glutathione peroxidase (GSH-PX) and superoxide dismutase (SOD) were determined using colorimetric methods with corresponding assay kits (Nanjing Jiancheng Bioengineering Institute, Nanjing, China). β-hydroxybutyrate (BHBA) was measured using a dehydrogenase method with a specific assay kit (Shanghai Zhangshi Biotech Co., Ltd., Shanghai, China).

### 2.6. Fecal Sample and URT Swab Collection

On the last day of the experiment, 10 g of rectal feces were collected in the morning after 2 h of feeding for the determination of fecal fermentation parameters. Meanwhile, the swabs were collected from the URT of calves by inserting a sterile 15 cm swab approximately 13 cm into the left nostril of each calf, reaching the nasopharynx, making contact with the mucosa and rotating 360° 3 times to obtain a URT swab, which was then folded and stored in 1.8 mL sterile polypropylene tubes in liquid nitrogen for analysis of the bacterial community of the URT.

### 2.7. Determination of SCFAs in Rectal Feces

For sample preparation, the rectal fecal samples were thawed at room temperature. Approximately 1.00 g of the sample was weighed and thoroughly mixed with 6 mL of ultrapure water. The mixture was then kept overnight at 4 °C. Afterward, the mixture was centrifuged at 3300 g/min for 10 min at 4 °C. Subsequently, 1 mL of the supernatant was transferred to a 1.5 mL centrifuge tube. To the centrifuge tube, 0.2 mL of a pyrophosphoric acid solution containing 2-ethylbutyric acid as an internal standard was added (prepared by weighing 25 g of pyrophosphoric acid and 0.217 mL of 2-ethylbutyric acid, and then adjusting the volume to 100 mL with ultrapure water to obtain a 25% (*w*/*v*) pyrophosphoric acid solution containing 2 g/L of 2-ethylbutyric acid). The mixture was thoroughly mixed and placed in an ice-water bath for 30 min, followed by centrifugation at 9500 g/min for 10 min at 4 °C. After centrifugation, 1 mL of the supernatant was collected and filtered through a 0.22 μM membrane into a vial for further analysis.

For standard solution preparation, 330 μL of acetic acid, 400 μL of propionic acid, 30 μL of isobutyric acid, 160 μL of butyric acid, 40 μL of isovaleric acid and 50 μL of valeric acid were added to a 100 mL volumetric flask. The volume was then adjusted to 100 mL with ultrapure water to prepare the mixed standard stock solution. Subsequently, 0.2 mL, 0.15 mL, 0.1 mL, 0.05 mL and 0.025 mL of the mixed standard stock solution were separately transferred to 1.5 mL centrifuge tubes. To the respective centrifuge tubes, 0 mL, 0.05 mL, 0.1 mL, 0.15 mL and 0.175 mL of ultrapure water were added to prepare 5 different gradient standard solutions. In each centrifuge tube, 0.2 mL of a pyrophosphoric acid solution containing 2-ethylbutyric acid as an internal standard was added. The mixture was thoroughly mixed and filtered through a 0.22 μM membrane into a vial for further analysis.

The detection was performed using an Agilent-7890 gas chromatograph with the following parameters: a 60 m × 0.25 mm × 0.50 μM DB-FFAP column was selected as the stationary phase. High-purity nitrogen gas (99.999%) was used as the carrier gas at a flow rate of 0.8 mL/min, and high-purity hydrogen gas (99.999%) was used as the auxiliary gas. The flame ionization detector (FID) temperature was set at 250 °C, while the injection port temperature was maintained at 220 °C. The split ratio was set at 40:1, and the injection volume was 1.5 μL. The temperature program included an initial temperature of 60 °C, which was ramped up at a rate of 20 °C/min to 120 °C and held for 3 min. Subsequently, the temperature was increased at a rate of 4 °C/min to 180 °C and held for 3 min.

### 2.8. URT Microbiome Analysis

#### 2.8.1. DNA Extraction

The chloroform-CTAB method was used to extract DNA from swab samples, following Arseneau [18] with minor modifications. Briefly, sterile steel beads and 100 μL CTAB buffer (2 g 2% CTAB, 28 mL 5 M NaCl, 10 mL 1 M Tris-HCL mixed with 4 mL 0.5 M EDTA and fixed to 100 mL with DNase-free water) were added to the sample tube and ground at 25 HZ for 7 min using an automated tissue disruptor (QIAGEN, Venlo, The Netherlands). We then added 50 μL of lysozyme and 5 μL of RNaseA and incubated the solution for 20 min at 37 °C. A total of 16 μL of proteinase K was then added and incubated for 20 min at 37 °C. Then, 25 μL of 20% SDS was added, mixed, lightly inverted and incubated for 1 h at 56 °C. We then added an equal volume of phenol:chloroform:isoamyl alcohol (25:24:1), mixed and inverted it then centrifuged it at 14,000 g/min for 15 min at 4 °C to denature and precipitate the proteins. The supernatant was transferred to a centrifuge tube containing 2/3 volume of isopropanol, mixed upside down and left to precipitate DNA in an ice bath for 30 min. The supernatant was discarded by centrifugation at 14,000 g/min for 15 min at 4 °C and washed twice with 75% ethanol. We added 100 μL of DNase-free water to resuspend the DNA. The quantity and quality of extracted DNA was measured using a NanoDrop NC2000 spectrophotometer (Thermo Fisher Scientific, Waltham, MA, USA) and agarose gel electrophoresis, respectively.

#### 2.8.2. 16S rRNA Gene Sequencing

The V3–V4 region of the bacterial 16S rRNA gene was amplified using the forward primer 338F (5′-ACTCCTACGGGAGGCAGCA-3′) and the reverse primer 806R (5′-GGACTACHVGGGTWTCTAAT-3′). The PCR reaction mixture consisted of 5 μL of buffer (5×), 0.25 μL of Fast pfu DNA polymerase (5 U/μL), 2 μL of dNTPs (2.5 mM), 1 μL each of forward and reverse primers (10 μM), 1 μL of DNA template and 14.75 μL of ddH2O. The PCR conditions included an initial denaturation step at 98 °C for 5 min, followed by 30 cycles of denaturation at 98 °C for 30 s, annealing at 53 °C for 30 s, and extension at 72 °C for 45 s. A final extension step was performed at 72 °C for 5 min. The PCR products were then subjected to agarose gel electrophoresis, followed by purification with magnetic beads. The purified PCR products were quantified using the Quant-iT PicoGreen dsDNA assay kit (Thermo Fisher Scientific, USA) and a microplate reader (BioTek, FLx800, Winooski, VT, USA). Based on the fluorescence quantification results, the samples were mixed in the appropriate ratios to meet the sequencing requirements of each sample. The sequencing libraries were constructed using the NovaSeq 6000 SP Reagent Kit (500 cycles) and sequenced on the Illumina NovaSeq platform (Illumina Inc., San Diego, CA, USA) at Panomics Biomedical Technology Co., Ltd., (Suzhou, China) with paired-end 2 × 250 bp sequencing.

### 2.9. Statistical Analysis

Each calf was considered as a statistical unit. The experimental data were initially organized using Microsoft Excel 2016 software (Microsoft, Redmond, WA, USA). Subsequently, the general linear model (GLM) procedure of SAS 9.4 software (SAS Institute Inc., North Carolina, USA) was employed to perform a 1-way analysis of variance (ANOVA). Multiple comparisons were conducted using the Tukey method, with differences considered significant when *p* < 0.05, highly significant when *p* < 0.01, and a significant trend observed when 0.05 < *p* < 0.1.

## 3. Results

### 3.1. Health Parameters

There was no death or fever or depression in the test calves during the entire study period. As shown in Table 2, the addition of CAP significantly decreased the total score (*p* = 0.01) and cough score (*p* = 0.04) in a dose-dependent manner, with group H being significantly lower than group C. There was a tendency for CAP to reduce rectal temperature score (*p* = 0.08), but there was no significant effect on nose score (*p* = 0.64) and eye and ear score (*p* = 0.90).

### 3.2. Serum Biochemical Indices

As shown in Table 3, the addition of CAP significantly decreased serum GLU concentration (*p* = 0.04) and tended to increase serum BUN concentration (*p* = 0.07). There were no significant differences in serum ALT, AST, TP, ALB, GLOB, ALP, CR and BHBA concentrations between the three groups (*p* > 0.05).

### 3.3. Serum Antioxidant Indices

Throughout the study period, addition of CAP significantly decreased serum GSH-PX concentration (*p* < 0.01; Table 4) in a dose-dependent manner. In addition, CAP also increased serum SOD concentration (*p* < 0.01) and decreased MDA concentration (*p* < 0.01). However, there was no significant difference in T-AOC concentration between the groups (*p* > 0.05).

### 3.4. Serum Immunity Indices

As shown in Table 5, CAP addition increased the concentration of three immunoglobulins, IgA (*p* = 0.03), IgG (*p* < 0.01) and IgM (*p* = 0.02), in a dose-dependent manner, and increased IL-10 concentration (*p* = 0.03). In addition, addition of CAP decreased serum IL-1β (*p* = 0.04), IL-6 (*p* = 0.02), SAA (*p* = 0.02) and HP (*p* < 0.01) concentrations but had no significant effect on TNF-α concentrations (*p* > 0.05).

### 3.5. Fecal Fermentation Parameters

As shown in Table 6, the addition of CAP increased the fecal total SCFA (*p* = 0.02), acetic acid (*p* < 0.01) and butyric acid (*p* < 0.01) concentrations and had no significant effect on propionic acid, isobutyric acid, isovaleric acid and valeric acid (*p* > 0.05).

### 3.6. URT Microbial Composition

#### 3.6.1. Alpha Diversity Analysis and Beta Diversity Analysis

The alpha diversity analysis showed that the Chao1 index was significantly different among the three groups (*p* = 0.03, Figure 1A). Observed species and Shannon and Simpson indices were not significantly different among the three groups. The beta diversity analysis based on UniFrac’s weighted PCoA as well as Adonis difference analysis showed significant differences in URT microbial composition among groups of calves (*p* = 0.001; Figure 1B).

#### 3.6.2. Relative Abundance and Structure of URT Microbiota at Phylum and Genus Levels

The effect of CAP on the URT microbiota of calves was further assessed at the phylum and genus levels. The results showed that at the phylum level, several groups with larger mean relative abundance (MRA) were different in all three groups, but mainly *Proteobacteria*, *Tenericutes*, *Bacteroidetes*, *Actinobacteria* and *Firmicutes* (>97%; Figure 2A). Among these groups, *Proteobacteria* (*p* < 0.01, Table 7), *Tenericutes* (*p* < 0.01) and *Bacteroidetes* (*p* = 0.01) were significantly different among the three groups. At the genus level, the predominant bacteria were *Mycoplasma*, *Moraxella* and *Mannheimia* (Figure 2B). Significant differences were found among the three groups for *Mycoplasma* (*p* < 0.01, Table 8), *Moraxella* (*p* < 0.01) and *Corynebacterium* (*p* < 0.01). The MRA of *Mycoplasma* and *Corynebacterium* in group H was 65.70% and 14.46% of that in group C. The MRA of *Mycoplasma* and *Corynebacterium* in group L was 148.43% and 0.85% of that in group C. The MRA of *Mycoplasma* and *Corynebacterium* in group L was 148.43% and 0.85% of that in group C. In addition, there was a tendency to reduce *Staphylococcus* in the CAP treatment group (*p* = 0.059). The MRA of *Staphylococcus* in the H group was 31.91% of that in the C group, and the MRA of *Staphylococcus* in the L group was 6.56% of that in the C group.

## 4. Discussion

Serum biochemical parameters are important indicators for assessing the health status and organ function of animals. In this study, the addition of CAP regulated GLU metabolism and decreased serum GLU concentration. As calves grow and develop, the energy source shifts from GLU to volatile fatty acids and GLU concentration gradually decreases [19]. Due to the underdeveloped rumen and immature digestive system of calves, the nutrients of the diet are mainly digested and absorbed in the gut [20]. SCFA is fermented by anaerobic microorganisms in the gut using carbohydrates such as resistant starch, cellulose, and oligosaccharides that are difficult for the body to digest, and its level in the gut is closely related to gut microbial activity. In addition to providing energy to host intestinal epithelial cells, SCFA also plays a role in regulating gut mucosal immunity, modulating microflora, improving gut function and antitumor activity [21]. In this study, the addition of CAP improved intestinal fermentation and increased the fecal total SCFA, acetic acid and butyric acid content. Among them, butyric acid not only provides part of the energy for the activity of gut epithelial cells but also stimulates the proliferation of epithelial cells and promotes the growth and development of the large intestine [22]. In a similar study, the addition of capsaicin to beef Holstein heifers improved rumen fermentation, lowered rumen pH and increased total VFA concentration [23]. In conclusion, CAP can regulate GLU metabolism and gut fermentation in calves, which is beneficial for healthy calf growth.

Group pen housing increases the risk of pathogen transmission among calves. Compared to individually housed calves, group-housed calves have a higher risk of BRD [3]. In this study, the addition of CAP decreased the overall respiratory system score, cough score, and showed a trend of reducing rectal temperature score, with the high dose of CAP exhibiting better effects. These findings suggest that the addition of CAP is beneficial for the healthy growth of calves under group housing conditions, reducing the incidence of BRD. Serum antioxidant markers are important indicators of an organism’s ability to maintain oxidative balance [24]. Antioxidant enzymes, such as GSH-PX, SOD and CAT, help alleviate the impact of oxidative stress, while MDA is a product of lipid peroxidation [10]. In this study, the addition of CAP increased activities of GSH-PX and SOD in serum while reducing serum MDA concentration. The phenolic hydroxyl structure of CAP possesses a strong free radical scavenging ability and can prevent lipid peroxidation and protein oxidation by eliminating peroxide radicals [25,26]. In vitro experiments have demonstrated that CAP can enhance the resistance of low-density lipoprotein to oxidation by delaying the onset of oxidation and slowing down the rate of oxidation [27]. In vivo experiments have shown that the addition of CAP reduces the serum total cholesterol and lipid peroxide concentration in rats [28].

Serum immunoglobulin concentration and inflammatory factors are important indicators of immune function in the body. In this study, the addition of CAP increased the concentration of three immunoglobulins (IgA, IgG and IgM) as well as IL-10, while reducing serum concentration of IL-1β and IL-6, with the high dose of CAP showing better effects. Similar studies have found that CAP may downregulate the expression of pro-inflammatory cytokines, such as TNF-α, IL-6 and IL-8, through the NF-κB signaling pathway, while upregulating the expression of the anti-inflammatory cytokine IL-10 [29,30]. CAP has been shown to enhance the immune capacity of rats by increasing IL-10 concentration and suppressing the expression of IL-6 cytokines [9].

SAA and HP are two major acute-phase proteins in cattle, and their concentrations increase when animals experience stress such as inflammation and infection [31]. In this study, the addition of CAP decreased serum concentration of SAA and HP, with the high dose of CAP showing better effects. Research has shown that calves with respiratory system diseases have significantly higher concentrations of pro-inflammatory cytokines, SAA and HP in their serum compared to healthy calves [32], which is consistent with our study results. In summary, the addition of CAP can enhance the antioxidant and immune capacities of calves, reduce the concentration of stress biomarkers and consequently lower the respiratory system score, promoting calf health.

URT is a critical window of entry for pathogens and a potential pathway for lower respiratory and middle ear infections with a complex microbial community composed of commensal microorganisms and potential pathogens [33]. Studies have reported that the main bacteria associated with BRD, *Mannheimia haemolytica*, *Pasteurella multocida*, *Mycoplasma bovis* and *Histophilus somni*, are prevalent in calves with URT infection [34]. Calves housed in groups have an increased susceptibility to BRD and a higher risk of disease transmission [34]. To investigate the effect of CAP on the URT microbiota, this study housed the same group of calves in the same large pen and sampled them for analysis at the end of the trial. The results showed that the chao1 index was significantly lower in the control group than in the CAP-treated group. The chao1 index can characterize microbial community richness. It was found that bacterial diversity and richness were significantly lower in the URT of calves diagnosed with BRD [35], similar to the results of the present study, which corresponds to a higher respiratory profile in the control group.

At the phylum level, the most abundant bacteria observed in each group were *Proteobacteria*, *Tenericutes*, *Bacteroidetes*, *Actinobacteria* and *Firmicutes*, similar to the findings of other studies on the URT microbiota of calves [34,36]. Feeding high doses of CAP significantly increased the abundance of *Proteobacteria*. This result occurred because high doses of CAP significantly increased the mean relative abundance of *Moraxella*. *Moraxella* has been found to be one of the most abundant genera in bovine URT in many studies [37,38]. However, the results of reports related to the relationship between *Moraxella* and BRD are inconsistent, with some studies reporting an association between *Moraxella* and the development of pneumonia and otitis media in dairy cattle [33,39]. However, McMullen et al. [40] found high abundance of *Moraxella* in the nostrils and nasopharynx of healthy cattle, which may indicate that some strains of *Moraxella* may be part of a normal, healthy nasal microbiota. The high abundance of *Moraxella* in group H may also be due to a reduction in *Mycoplasma* and *Mannheimia* resulting in decreased microbial competition in the URT. Feeding high doses of CAP also significantly decreased the abundance of *Tenericutes* due to the reduction in the mean relative abundance of *Mycoplasma* by high doses of CAP. *Mycoplasma bovis* is the main causative agent of BRD, and a study reported by Lima et al. [33] found significantly higher *Mycoplasma* abundance in the URT of calves with BRD than in healthy calves. This likewise corresponds to higher respiratory scores in the control group. However, low-dose CAP did not reduce the mean relative abundance of *Mycoplasma*, which could be due to dose differences in CAP. In addition, the addition of CAP decreased the abundance of *Bacteroidetes*. However, due to the short sequencing read length, specific genera of bacteria could not be identified at the genus level. The addition of CAP also decreased the MRA of *Corynebacterium* at the genus level and there was also a trend to reduce the MRA of *Staphylococcus*. *Corynebacterium Pseudotuberculosi* is one of the major causative agents of bovine otitis media. In addition, *Staphylococcus aureus* is an important causative bacterium of BRD [41], and mastitis caused by *Staphylococcus* is an important disease in dairy cows [42]. These data suggest that the addition of CAP can reduce the abundance of BRD-causing bacteria in URT and thus reduce BRD prevalence in a herd environment. However, the URT microbiota in calves has not been studied in sufficient depth and there are not enough references to adequately explain the current findings. Future studies could use whole-genome sequencing to analyze the role played by the URT microbiota in calf respiratory health.

## 5. Conclusions

In summary, high-dose CAP decreased calf respiratory scores, increased serum antioxidant enzymes, immunoglobulin and anti-inflammatory factor concentrations, and decreased serum peroxide and stress protein concentrations. High-dose CAP also increased rectal SCFA concentrations and decreased the MRA of harmful bacteria in the URT. Thus, a high dose of CAP is beneficial for calf health by increasing immune competence.

## Figures and Tables

**Figure 1 microorganisms-11-01903-f001:**
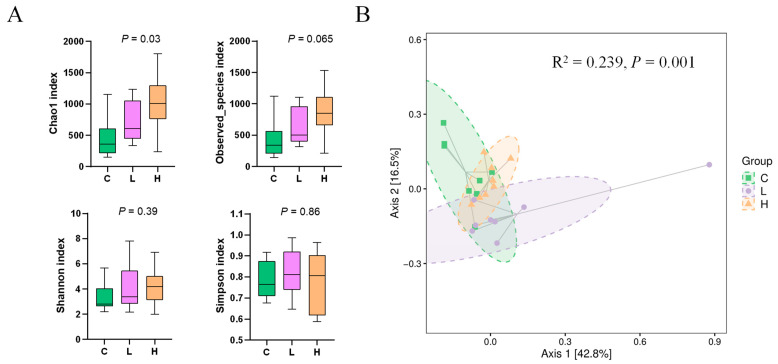
Alpha diversity indices (**A**) and weighted UniFrac-based PCoA plot (**B**) of the URT swab samples of Holstein calves fed the control diet (C) or supplemented diet with low-level capsaicin (L) or supplemented diet with high-level capsaicin (H).

**Figure 2 microorganisms-11-01903-f002:**
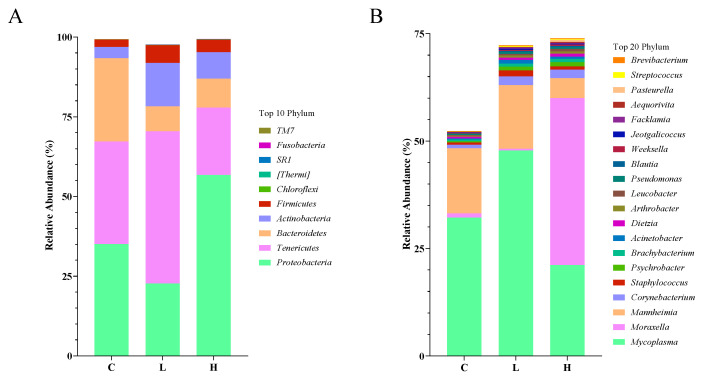
The composition of the URT microflora on the phylum (**A**) and genus (**B**) levels of the URT swab samples of Holstein calves fed the control diet (C) or supplemented diet with low-level capsaicin (L) or supplemented diet with high-level capsaicin (H).

**Table 1 microorganisms-11-01903-t001:** Chemical composition (% of DM unless otherwise noted) of milk replacer, calf starter and oat hay.

Item	Milk Replacer ^1^	Calf Starter ^2^	Oat Hay
DM (%)	94.8	88.5	95.1
Ash	9.1	9.6	7.6
CP	20.2	20.9	10.5
EE	16.2	5.3	2.3
NDF	-	21.9	58.7
ADF	-	11.0	35.5
NFC	54.5	42.3	20.9

^1^ Milk replacer (Nutrifeed, The Netherlands). ^2^ Commercial calf starter (Guangdong Wen’s Dairy Industry Co., Zhaoqing, China).

**Table 2 microorganisms-11-01903-t002:** Respiratory system score of Holstein calves fed the control diet (C) or supplemented diet with low-level capsaicin (L) or supplemented diet with high-level capsaicin (H) ^1^.

Item	Treatment			SEM	*p*-Value
	C	L	H		
Total score	4.80 ^a^	4.16 ^ab^	3.67 ^b^	0.14	0.01
Cough score	1.23 ^a^	0.94 ^ab^	0.66 ^b^	0.08	0.04
Nose score	1.20	1.15	1.01	0.08	0.64
Eye and Ear score	1.40	1.38	1.38	0.06	0.90
Rectal temperature score	0.98	0.69	0.60	0.07	0.08

^1^ The values (row) with unlike lowercase letters differed significantly (*p* < 0.05).

**Table 3 microorganisms-11-01903-t003:** Blood biochemical indices of Holstein calves fed the control diet (C) or supplemented diet with low-level capsaicin (L) or supplemented diet with high-level capsaicin (H) ^1^.

Item	Treatment			SEM	*p*-Value
	C	L	H		
GLU, mmol/L	9.03 ^a^	7.87 ^b^	7.69 ^b^	0.22	0.04
ALT, U/L	12.15	12.62	12.85	0.29	0.61
AST, U/L	55.20	48.87	54.28	2.10	0.43
TP, g/L	56.98	56.77	56.89	1.11	1.00
ALB, g/L	26.92	26.92	26.24	0.39	0.72
GLOB, g/L	30.06	29.85	30.65	1.11	0.95
ALP, U/L	244.29	231.14	241.77	23.0	0.97
BUN, mmol/L	2.80	3.57	3.28	0.13	0.07
CR, μmol/L	86.20	82.02	82.84	1.37	0.43
BHBA, mmol/L	0.14	0.18	0.19	0.01	0.10

^1^ The values (row) with unlike lowercase letters differed significantly (*p* < 0.05).

**Table 4 microorganisms-11-01903-t004:** Blood antioxidant indices of Holstein calves fed the control diet (C) or supplemented diet with low-level capsaicin (L) or supplemented diet with high-level capsaicin (H) ^1^.

Item	Treatment			SEM	*p*-Value
	C	L	H		
T-AOC, mmol/L	0.79	0.78	0.77	0.01	0.84
GSH-PX, U/mL	83.10 ^b^	100.07 ^ab^	118.99 ^a^	3.47	<0.01
SOD, U/mL	145.65 ^b^	156.92 ^a^	156.27 ^a^	1.41	<0.01
MDA-nmol/mL	4.51 ^a^	3.13 ^b^	3.74 ^ab^	0.14	<0.01

^1^ The values (row) with unlike lowercase letters differed significantly (*p* < 0.05).

**Table 5 microorganisms-11-01903-t005:** Blood immunity indices of Holstein calves fed the control diet (C) or supplemented diet with low-level capsaicin (L) or supplemented diet with high-level capsaicin (H) ^1^.

Item	Treatment			SEM	*p*-Value
	C	L	H		
IgA, μg/mL	1321.05 ^b^	1528.00 ^ab^	1766.45 ^a^	63.53	0.03
IgG, mg/mL	5.41 ^b^	6.24 ^ab^	6.66 ^a^	0.15	<0.01
IgM, μg/mL	1036.45 ^b^	1156.31 ^ab^	1329.40 ^a^	38.41	0.02
IL-1β, pg/mL	415.10 ^a^	412.78 ^a^	320.78 ^b^	15.78	0.04
IL-6, pg/mL	99.65 ^a^	96.99 ^a^	74.93 ^b^	3.55	0.02
IL-10, pg/mL	52.31 ^b^	53.62 ^b^	62.66 ^a^	1.59	0.03
TNF-α, pg/mL	90.02	85.36	64.75	4.87	0.10
SAA, μg/mL	6.14 ^a^	5.80 ^ab^	4.90 ^b^	0.17	0.02
HP, μg/mL	23.84 ^a^	20.27 ^ab^	18.19 ^b^	0.68	<0.01

^1^ The values (row) with unlike lowercase letters differed significantly (*p* < 0.05).

**Table 6 microorganisms-11-01903-t006:** Fecal SCFA content (mmoL/L) of Holstein calves fed the control diet (C) or supplemented diet with low-level capsaicin (L) or supplemented diet with high-level capsaicin (H) ^1^.

Item	Treatment			SEM	*p*-Value
	C	L	H		
Total SCFA	37.69 ^b^	39.94 ^ab^	57.07 ^a^	2.72	0.02
Acetic acid	24.42 ^b^	26.11 ^b^	36.79 ^a^	1.52	<0.01
Propionic acid	8.22	8.03	11.38	1.03	0.35
Butyric acid	3.03 ^b^	3.60 ^ab^	6.40 ^a^	0.37	<0.01
Isobutyric acid	0.74	0.63	1.02	0.09	0.22
Isovaleric acid	0.79	1.11	0.95	0.19	0.80
Valeric acid	0.48	0.46	0.53	0.12	0.97

^1^ The values (row) with unlike lowercase letters differed significantly (*p* < 0.05).

**Table 7 microorganisms-11-01903-t007:** Top five relative abundances of the URT microbiota at the phylum level of Holstein calves fed the control diet (C) or supplemented diet with low-level capsaicin (L) or supplemented diet with high-level capsaicin (H).

Item	Treatment			SEM	*p*-Value ^1^
	C	L	H		
*Proteobacteria*	35.04 ^b^	22.67 ^b^	56.73 ^a^	0.03	<0.01
*Tenericutes*	32.20 ^ab^	47.79 ^a^	21.16 ^b^	0.03	<0.01
*Bacteroidetes*	26.08 ^a^	7.82 ^b^	9.01 ^b^	0.03	0.01
*Actinobacteria*	3.56	13.59	8.28	0.02	0.18
*Firmicutes*	2.36	5.55	3.94	0.01	0.46

^1^ The values (row) with unlike lowercase letters differed significantly (*p* < 0.05).

**Table 8 microorganisms-11-01903-t008:** Top 10 relative abundances of the URT microbiota at the genus level of Holstein calves fed the control diet (C) or supplemented diet with low-level capsaicin (L) or supplemented diet with high-level capsaicin (H).

Item	Treatment			SEM	*p*-Value ^1^
	C	L	H		
*Mycoplasma*	32.19 ^ab^	47.78 ^a^	21.15 ^b^	0.03	<0.01
*Moraxella*	0.96 ^b^	0.41 ^b^	38.88 ^a^	0.03	<0.01
*Mannheimia*	15.2	14.81	4.60	0.03	0.33
*Corynebacterium*	20.06 ^a^	0.17 ^b^	2.90 ^b^	0.02	<0.01
*Staphylococcus*	13.10	0.86	4.18	0.02	0.06
*Psychrobacter*	4.77	5.48	3.55	0.02	0.90
*Brachybacterium*	0.54	7.85	1.81	0.02	0.32
*Acinetobacter*	3.74	1.52	3.49	0.01	0.66
*Dietzia*	0.81	2.06	1.96	0.00	0.33
*Arthrobacter*	0.53	1.39	0.78	0.00	0.34

^1^ The values (row) with unlike lowercase letters differed significantly (*p* < 0.05).

## Data Availability

The raw sequence data reported in this paper have been deposited in the Genome Sequence Archive (Genomics, Proteomics and Bioinformatics 2021) in the National Genomics Data Center (Nucleic Acids Res 2022), China National Center for Bioinformation/Beijing Institute of Genomics, Chinese Academy of Sciences (GSA: CRA011263) that are publicly accessible at https://ngdc.cncb.ac.cn/gsa/ (accessed on 2 June 2023).

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
