# Peer review of "Effect of Capsaicin Addition on Antioxidant Capacity, Immune Performance and Upper Respiratory Microbiota in Nursing Calves"

_microorganisms, 2023, doi:10.3390/microorganisms11081903_

Round 1

Reviewer 1 Report

line 57:  Insertion of an hypothesis before aim statement seems advisable.

line 69 and 77:  Unpublished study and previous study are confusing.   Please clarify.

line 88 and 93:  Reconstruct sentences so that numerals do not initiate the sentence.

line 95:  Because of your feeding system, there is no assurance that each calf received the stated dosage of CAP.  Any comment?

lines 97-100:  Dry matter determination is confusing.  Rehydration step?

line 131:  Convert rpm to x g here and throughout the manuscript.

line 149:  Define URT.

line 192:  Chlorine is Cl and not CL.

line 252:  Change reduced to decreased and level to concentration here and elsewhere in manuscript.

Table 6:  Change ml to mL.  Insert in blood after indices in title to table.

line 279:  Delete had no significant effect from end of sentence.

Figure 2:  Axis label and phylum and genus names need enlarging if possible for east of reading.

line 333:  Do you really mean growth performance of dairy cows?

line 334:  What is a nursing cow?  Do you mean lactating cow?

line 338:  different experimental animals in an incomplete sentence.

line 339 and 340:  Change calf intake to intake of calves.

line 343:  Define BRD.

line 348:  New paragraph?

line 351:  What is a serum level?  Change to increased activities of GSH-PX and SOD in serum.

line 379:  Insert infection after URT.

line 419:  Delete at lower levels at end of sentence.

line 421:  Change calve to calf.

line 425:  Consider adding an overall concluding sentence here.

line 358:  New paragraph?

Reviewer 2 Report

This is an interesting study showing potential value of CAP to reduce BRD in calves. However, there are various components of this study that has not been well described. In a very small experiment, such as this one, with only 8 calves per group, you need to describe the change within the calf before and after treatment. Variables such as serum Ig levels can not be presented without taking into account the serum Ig levels at the start of the experiment. It seems that you have data of the calves from birth day, thus use this data to control for pre-experiment levels within the calf. In a study with only 24 calves, you should not present p-values with precision to the 1/1000 decimal, two decimals is sufficient. Please explain power calculations for the study.
